

# Intertwining orbital current order and superconductivity in kagome metal

Hyeok-Jun Yang[1], Hee Seung Kim[1], Min Yong Jeong[1],
Yong Baek Kim[2*], Myung Joon Han[1†] and SungBin Lee[1‡]

**1** Department of Physics, Korea Advanced Institute
of Science and Technology, Daejeon, 34141, Korea
**2** Department of Physics and Centre for Quantum Materials,
University of Toronto, Toronto, Ontario M5S 1A7, Canada

★ ybkim@physics.utoronto.ca, † m.j.han@kaist.ac.kr, ‡ sungbin@kaist.ac.kr

## Abstract

The nature of superconductivity in newly discovered kagome materials, $AV_3Sb_5$ (A = K, Rb, Cs), has been a subject of intense debate. Recent experiments suggest the presence of orbital current order on top of the charge density wave (CDW) and superconductivity. Since the orbital current order breaks time-reversal symmetry, it may fundamentally affect possible superconducting states. In this work, we investigate the mutual influence between the orbital current order and superconductivity in kagome metal with characteristic van Hove singularity (vHS). By explicitly deriving the Landau-Ginzburg theory, we classify possible orbital current order and superconductivity. It turns out that distinct unconventional superconductivities are expected, depending on the orbital current ordering types. Thus, this information can be used to infer the superconducting order parameter when the orbital current order is identified and vice versa. We also discuss possible experiments that may distinguish such superconducting states coexisting with the orbital current order.



# 1   Introduction

Itinerant electron systems on the kagome lattice has long been considered as a fertile ground for exotic quantum ground states due to the presence of flat band, van Hove singularity (vHS), and non-trivial band topology [1–11]. The recent discovery of kagome materials, $AV_3Sb_5$ (A=K, Rb, Cs) has provided an excellent platform to explore such emergent collective phases in full glory [12]. Particularly, most attention has been paid to unconventional charge density wave (CDW) with a large unit-cell, and the subsequent appearance of superconductivity (SC) at much lower temperature [12, 12–25, 25–33].

Interestingly, the anomalous Hall effect was observed below the CDW transition temperature [27, 29] while no static magnetic order was detected in the neutron scattering and muon spectroscopy [12, 25, 28]. A prime candidate is the orbital current order, which can be characterized by an imaginary version of the CDW (iCDW) order parameter [19, 24, 34–38]. In this case, the spin degrees of freedom plays an important role in determining the iCDW type as distinct symmetry breaking patterns. Moreover, in the superconducting state, the orbital current order and CDW may coexist [18, 20, 25, 39]. It is therefore likely that understanding the orbital current order with spin degrees of freedom is essential to clarify the superconducting order parameter.

In this work, we focus on intertwining relationships between the spin-dependent orbital current order and superconductivity in the kagome metal with characteristic vHS. Based on the Landau-Ginzburg (LG) theory, we establish the relationship between different orbital current orders and superconducting order parameters when their critical temperatures are energetically close. Considering the spin degrees of freedom and three vHS of the kagome lattice, we find that there are four possible orbital current orders and they are closely related to four different SC order parameters. It turns out that the orbital current order is closely related to both the non-trivial band topology and unconventional SC order parameters. We focus on how these orbital current and superconducting orders are intertwined based on symmetry properties rather than the material specification where a variety of collective orders are involved. In this case, the identification of the orbital current order would imply the emergence of a particular kind of superconducting order and vice versa. We discuss the scanning tunneling microscopy (STM) and Josephson junction experiments that may be able to distinguish different orbital current orders and superconducting states. Our study provides a way to understand the correlation between orbital current order and superconductivity in various kagome metals.

**Sci**|**Post**                                                   SciPost Phys. Core 6, 008 (2023)

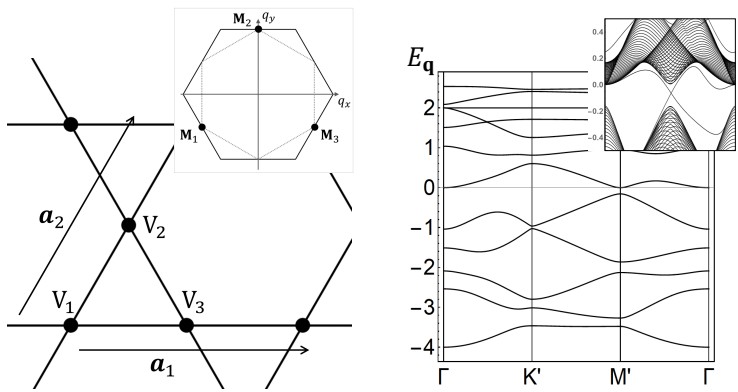

Figure 1: Left panel: The kagome lattice, where the Vanadium sits on sublattice sites $V_{\alpha=1,2,3}$ with the primitive lattice vector $\mathbf{a}_1 = (1,0)$ and $\mathbf{a}_2 = (\frac{1}{2}, \frac{\sqrt{3}}{2})$ in unit of lattice constant $a = 1$. (Inset: The BZ with vHS points, $\mathbf{M}_\alpha$.) Right panel: Chiral flux phase band structure for $t = 1, |\Phi_{\alpha\sigma}| = 0.3$. Due to the extended $2 \times 2$ unit cell, the BZ is folded with 12 bands per spin. Here, $K' = (\frac{2\pi}{3}, 0)$, $M' = (\frac{\pi}{2}, \frac{\pi}{2\sqrt{3}})$. (Inset: The edge spectrum near $E_{\mathbf{q}} = 0$ for case (i) in Eq. (5).)

## 2 Patch model

Close to the vHS, the electronic structure of AV$_3$Sb$_5$ is described by the $d$-orbitals on the Vanadium sites, which form a kagome lattice structure shown in Fig. 1 [12, 18, 21, 24, 30, 31]. Focusing on the vHS, we consider the case where the vHS near the Fermi level is mainly described by a single $d$-orbital, for simplicity.

Starting from the nearest-neighbour tight-binding model, there are three inequivalent vHS points in the Brillouin zone (BZ).

$$\mathbf{M}_1 = (-\pi, -\frac{\pi}{\sqrt{3}}), \quad \mathbf{M}_2 = (0, \frac{2\pi}{\sqrt{3}}), \quad \mathbf{M}_3 = (\pi, -\frac{\pi}{\sqrt{3}}). \tag{1}$$

For such vHS, the logarithmic divergence of the density of states (DOS) makes the low-energy physics mainly dominated by excitations near the saddle points [38, 40]. Thus, we adopt the patch model by defining the electronic fields $\psi_{\mathbf{k}\sigma} = (c_{1\mathbf{k}\sigma}, c_{2\mathbf{k}\sigma}, c_{3\mathbf{k}\sigma})$ close to $\mathbf{M}_{\alpha=1,2,3}$. Then, in the continuum limit, the dispersions near the vHS are represented as,

$$\epsilon_{1,\mathbf{k}} = \frac{t}{2}k_x(k_x + \sqrt{3}k_y), \quad \epsilon_{2,\mathbf{k}} = -\frac{t}{4}(k_x^2 - 3k_y^2), \quad \epsilon_{3,\mathbf{k}} = \frac{t}{2}k_x(k_x - \sqrt{3}k_y), \tag{2}$$

where $\mathbf{k} \equiv \mathbf{q} - \mathbf{M}_\alpha$ is the momentum deviation from each vHS and is restricted to be inside a patch with a finite size $\Lambda$ around $\mathbf{M}_\alpha$. Here, $t$ is the nearest-neighbour hopping parameter. This simplification enables us to focus on the instabilities, concerning the particle-hole and pair condensates only at $\mathbf{Q} = \mathbf{0}$ or $\mathbf{Q} = \mathbf{M}_\alpha$ by integrating out $\psi_{\mathbf{k}\sigma}$-fields.

## 3 iCDW patterns

To consider the orbital current order, we focus on the case where the CDW order parameters are pure imaginary, so labelled as iCDW. Denoting the interaction strength responsible for

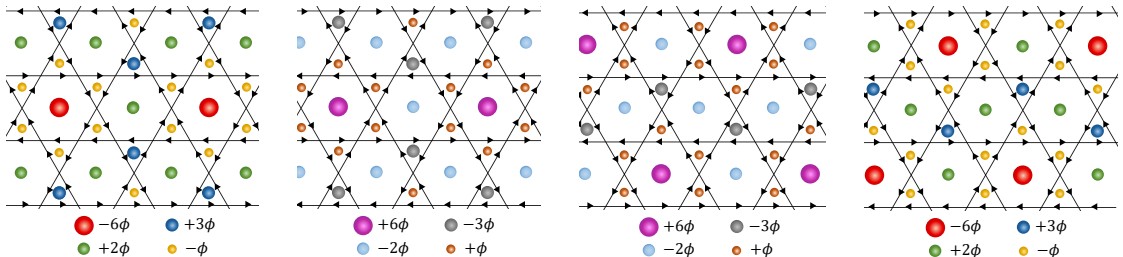

Figure 2: The down-spin $3Q$-iCDW patterns for 4 cases (a) $(\Phi_{1\downarrow}, \Phi_{2\downarrow}, \Phi_{3\downarrow}) = (i, i, i)$, $(\Phi_{1\downarrow}, \Phi_{2\downarrow}, \Phi_{3\downarrow}) = (-i, -i, -i)$, $(\Phi_{1\downarrow}, \Phi_{2\downarrow}, \Phi_{3\downarrow}) = (-i, i, i)$, and $(\Phi_{1\downarrow}, \Phi_{2\downarrow}, \Phi_{3\downarrow}) = (-i, -i, i)$. (from left to right, see Eq. (5) and Table. 1 in the main text.) The arrow and filled circle mark the bond current and the plaquette flux respectively. Here, $\phi$ is the Peierls phase along the nearest-neighbour hopping.

iCDW instability as $I_{\text{iCDW}}$, the iCDW order parameter is represented as [37, 41],

$$\Phi_{\alpha\sigma} = \frac{|I_{\text{iCDW}}|}{\Lambda^2} \sum_{\substack{|\mathbf{k}|<\Lambda \\ \alpha\beta\gamma}} \epsilon_{\alpha\beta\gamma} \langle c^\dagger_{\gamma\mathbf{k}\sigma} c_{\beta\mathbf{k}\sigma} \rangle, \tag{3}$$

where $\{\alpha, \beta, \gamma\} \in \{1, 2, 3\}$ are sublattice indices. In the kagome structure with a single orbital model, each vHS is composed of independent sublattice [4, 37, 42] and thus, the iCDW is directly proportional to the bond current flowing from $V_\beta$ site to $V_\gamma$ site.

Motivated by experimental observations [15, 16, 19, 22, 43], we now focus on $3Q$-iCDW states i.e., $\Phi_{1,2,3,\uparrow,\downarrow} \neq 0$. Their stability compared to $1Q$- and $2Q$-iCDW will be discussed later. Here, we consider the equal amplitudes for $\Phi_{1,2,3,\uparrow,\downarrow}$ which is shown later to minimize the LG free energy. Without loss of generality, we first fix the iCDW order parameters for spin-up electrons $\Phi_\uparrow \equiv (\Phi_{1\uparrow}, \Phi_{2\uparrow}, \Phi_{3\uparrow}) = |\Phi_{1\uparrow}|(i, i, i)$. One can easily check that the other cases are related by an inversion $\mathcal{I}$ at the lattice site, or a mirror reflection $\mathcal{M}$ along the lattice plane.

$$\begin{aligned} \mathcal{I}: \quad & (\Phi_{1\sigma}, \Phi_{2\sigma}, \Phi_{3\sigma}) \rightarrow (-\Phi_{1\sigma}, -\Phi_{2\sigma}, \Phi_{3\sigma}), \\ \mathcal{M}: \quad & (\Phi_{1\sigma}, \Phi_{2\sigma}, \Phi_{3\sigma}) \rightarrow (-\Phi_{1\sigma}, -\Phi_{2\sigma}, -\Phi_{3\sigma}). \end{aligned} \tag{4}$$

We note that $\mathcal{I}$ preserves the Chern number while $\mathcal{M}$ reverses it. Given the parameter $\{\Phi_{\alpha\uparrow}\}$, there are 4 independent patterns of $\Phi_\downarrow = (\Phi_{1\downarrow}, \Phi_{2\downarrow}, \Phi_{3\downarrow})$,

$$\begin{aligned} &\text{(i)} \ \ \Phi_\downarrow = (i, i, i), \qquad &&\text{(ii)} \ \ \Phi_\downarrow = (-i, -i, -i), \\ &\text{(iii)} \ \ \Phi_\downarrow = (-i, i, i), \qquad &&\text{(iv)} \ \ \Phi_\downarrow = (-i, -i, i), \end{aligned} \tag{5}$$

up to the 3-fold rotation (the amplitude is omitted for brevity). In Fig. 2, we exhibit the Peierls phases and the plaquette fluxes for 4 iCDW patterns. Unlike the spinless case, the 4 patterns in Eq. (5) are differentiated by the relative bond currents of spin-up and spin-down electrons and are not related by any symmetries with each other. Only the case (ii) preserves $\mathcal{T}$-symmetry, $\Phi^*_{\alpha\uparrow} = \Phi_{\alpha\downarrow}$, and the others break $\mathcal{T}$-symmetry. Since the iCDW order preserves the spin polarization, the spin-up and spin-down electronic bands are degenerate as shown in Fig. 1. The degeneracy of 4 patterns is not an artifact of our model, rather it is a consequence of symmetry properties Eq. (4) in the bulk.

The physical quantity characterizing each case in Eq. (5), is the topological invariant, the Chern number. For spin-up electrons, the Chern number is obtained for the bands occupied below the chemical potential $\mu = 0$ and the total Chern number turns out to be $\mathcal{C}^\uparrow_{xy} = +1$ [34], implying a spin-up edge mode.

Table 1: Symmetry properties of 4 possible iCDW phases in Eq. (5) with the spin resolved total Chern numbers. Only the down-spin order parameters are shown below while the reference up-spin components are given by $(\Phi_{1\uparrow}, \Phi_{2\uparrow}, \Phi_{3\uparrow}) = (i, i, i)$. The mark O or × indicates whether the iCDW phase preserves the corresponding symmetry or not.

| $(\Phi_{1\downarrow}, \Phi_{2\downarrow}, \Phi_{3\downarrow})$ | $\mathcal{T}$ | $(\mathcal{T}\times\mathcal{I})$ | $(\mathcal{T}\times\mathcal{I}\times\mathcal{M})$ | $\mathcal{C}_{xy}^{\uparrow}$ | $\mathcal{C}_{xy}^{\downarrow}$ |
|---|---|---|---|---|---|
| (i) $(i, i, i)$ | × | × | × | +1 | +1 |
| (ii) $(-i, -i, -i)$ | O | × | × | +1 | −1 |
| (iii) $(-i, i, i)$ | × | O | × | +1 | −1 |
| (iv) $(-i, -i, i)$ | × | × | O | +1 | +1 |

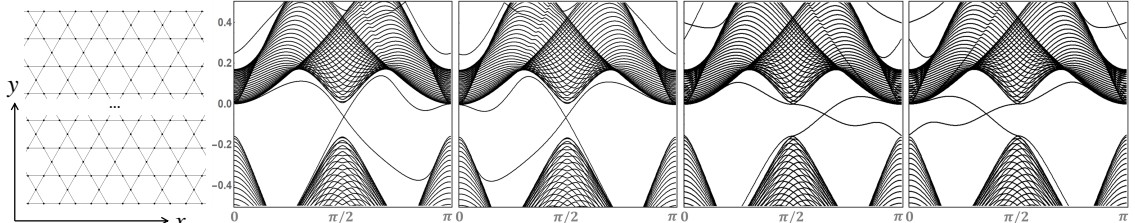

Figure 3: Left panel: The kagome lattice with an open boundary condition in the $y$-direction. The edge boundaries are cut on the upper and lower boundaries of the extended $2 \times 2$ unit cell. The down-spin $3Q$-iCDW edge spectra as a function of $k_x \in [0, \pi)$ for $(\Phi_{1\downarrow}, \Phi_{2\downarrow}, \Phi_{3\downarrow}) = (i, i, i)$, $(\Phi_{1\downarrow}, \Phi_{2\downarrow}, \Phi_{3\downarrow}) = (-i, -i, -i)$, $(\Phi_{1\downarrow}, \Phi_{2\downarrow}, \Phi_{3\downarrow}) = (-i, i, i)$, and $(\Phi_{1\downarrow}, \Phi_{2\downarrow}, \Phi_{3\downarrow}) = (-i, -i, i)$ (from left to right) with $|\Phi_{\alpha\downarrow}| = 0.3$.

The physical contents of the cases (i) and (ii) are rather obvious: case (i) is for the chiral flux phase with broken $\mathcal{T}$ along all bonds $\Phi_{\alpha\uparrow} = \Phi_{\alpha\downarrow}$ having $\mathcal{C}_{xy}^{\downarrow} = +1$, and case (ii) is for the helical phase with preserved $\mathcal{T}$ having $\mathcal{C}_{xy}^{\downarrow} = -1$. Although both cases of (iii) and (iv) break $\mathcal{T}$, they preserve $\mathcal{T} \times \mathcal{I}$ and $\mathcal{T} \times \mathcal{I} \times \mathcal{M}$-symmetries respectively. As a result, the corresponding Chern number is $\mathcal{C}_{xy}^{\downarrow} = -1$ and $+1$ for (iii) and (iv), respectively. These results are summarized in Table. 1.

Here, we exhibit the edge spectrum and spectral functions for all 4 cases. On the kagome lattice with an open boundary condition, the edge mode at zero energy manifests the nonzero Chern numbers in the bulk band structure. For down-spin electrons, the edge spectra for 4 cases are shown for $|\Phi_{\alpha\sigma}| = 0.3$ in Fig. 3.

## 4 Landau-Ginzburg theory

### 4.1 iCDW free energy

We now derive the LG free energy up to the quartic order in the presence of iCDW and SC order parameters. By integrating out $\psi_{\mathbf{k}\sigma}$-fields, we first examine the free energy of iCDW phase, $f_{\text{iCDW}} = f_{\text{iCDW}}^{\uparrow} + f_{\text{iCDW}}^{\downarrow}$ in terms of $\Phi_{\alpha\sigma}$ (Eq. (3)),

$$f_{\text{iCDW}}^{\sigma} = a(T - T_{\text{iCDW}})\Big(\sum_{\alpha} |\Phi_{\alpha\sigma}|^2\Big) + \frac{1}{2}u_1\Big(\sum_{\alpha} |\Phi_{\alpha\sigma}|^2\Big)^2 - (u_1 - u_2)\sum_{\alpha<\beta} |\Phi_{\alpha\sigma}|^2|\Phi_{\beta\sigma}|^2. \quad (6)$$

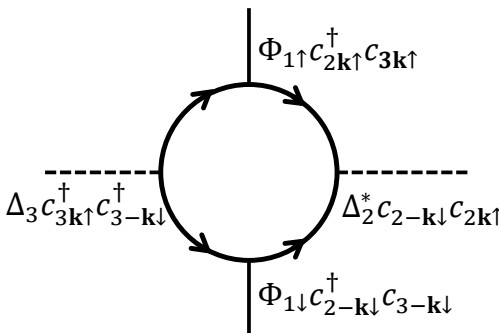

Figure 4: The diagrammatic representation for $u_4$-term in $f_{\text{CDW\&SC}}$ (Eq. (10)).

Here $a > 0$ is a constant and the cubic term, $\Phi_{1\sigma}\Phi_{2\sigma}\Phi_{3\sigma}$ + c.c. vanishes in the free energy since $\Phi_{\alpha\sigma}$ is purely imaginary. The iCDW order parameters $\Phi_{\alpha\sigma}$ (totally 6) are independent and their magnitudes in equilibrium are determined by minimizing the free energies Eq. (6). The inclusion of conventional CDW might promote 3$Q$-complex CDW or evenly the nematic charge order [36] but these possibilities are not considered here.

At low temperature $T \ll t\Lambda^2$, the quartic coefficients based on the dispersions in Eq. (2) are evaluated as,

$$u_1 \approx \frac{0.003}{tT^2}\log\Big(\frac{t\Lambda^2}{T}\Big), \qquad u_2 \approx \frac{0.006}{tT^2}, \tag{7}$$

up to the leading order [44]. In Appendix A, we explicitly derive Eq. (7) for the perfect nesting. Here, the condition $u_1 \gg u_2$ guarantees the stability of 3$Q$-iCDW state over 1$Q$- and 2$Q$-states, which is consistent with the experiments [15, 16, 19, 22, 43].

## 4.2 Total free energy

Now we include the SC order parameters. In the perfect nesting, Eq. (2), since the particle-particle susceptibility on the same patch is predominant [40], we focus on the SC order parameters which pair intra-patch excitations. Considering the singlet SC order parameters,

$$\Delta_\alpha = \frac{|I_{\text{SC}}|}{\Lambda^2} \sum_{|\mathbf{k}| < \Lambda} \langle c_{\alpha\mathbf{k}\downarrow} c_{\alpha-\mathbf{k}\uparrow} \rangle, \tag{8}$$

the total free energy is $f_{\text{total}} = f_{\text{iCDW}} + f_{\text{SC}} + f_{\text{CDW\&SC}}$ where

$$f_{\text{SC}} = a'(T - T_{\text{SC}})\Big(\sum_\alpha |\Delta_\alpha|^2\Big) + \frac{1}{2}u_3\Big(\sum_\alpha |\Delta_\alpha|^2\Big)^2 - u_3 \sum_{\alpha < \beta} |\Delta_\alpha|^2 |\Delta_\beta|^2, \tag{9}$$

and

$$f_{\text{CDW\&SC}} = \sum_{(\alpha\beta\gamma)} \Big[ -u_4\big(\Phi_{\alpha\uparrow}\Phi_{\alpha\downarrow}\Delta_\beta^*\Delta_\gamma + \text{c.c.}\big) + u_5\big(|\Phi_{\alpha\uparrow}|^2 + |\Phi_{\alpha\downarrow}|^2\big)\big(|\Delta_\beta|^2 + |\Delta_\gamma|^2\big) \Big]. \tag{10}$$

Here, the summation runs over the even permutations, $(\alpha\beta\gamma) = (1,2,3),(2,3,1),(3,1,2)$. We iterate that the free energies Eqs. (6), (9), (10) are formalized based on the symmetry properties of order parameters while the coefficients $u_1 \sim u_5$ are evaluated as functions of microscopic details such as $t, \Lambda$ and $T$.

Similar to the iCDW case in Eq. (6), the last term in Eq. (9) prefers the finite amplitude for the SC order parameters at different patches, namely $\Delta_\alpha \neq 0$ for all $\alpha$. In this case, the coupling terms between the SC order parameters at different patches prefer the same magnitudes and the pairing symmetry is determined by the relative phases $\text{Arg}(\Delta_\alpha)$. The irreducible channels of $\boldsymbol{\Delta}_{\text{patch}} = (\Delta_1, \Delta_2, \Delta_3)$ correspond to [40],

$$\boldsymbol{\Delta}_s = \sqrt{\frac{1}{3}}(1,1,1), \quad \boldsymbol{\Delta}_{d_{x^2-y^2}} = \sqrt{\frac{2}{3}}(1,-\frac{1}{2},-\frac{1}{2}), \quad \boldsymbol{\Delta}_{d_{xy}} = \sqrt{\frac{1}{2}}(0,1,-1), \quad (11)$$

so we consider $s$-wave or $d$-wave pairings and their mixtures. In Appendix B, the gap function in the microscopic Hamiltonian is formulated whose evaluation at vHS reduces to the patch model with Eq. (11).

The first term in Eq. (10) might be attractive depending on the phase relationship between iCDW and SC while the second term is always repulsive ($u_5 > 0$). Likewise, in Eq. (7), these coefficients can be evaluated and turn out to be $u_1 \approx u_3 \approx u_4 \approx u_5$ up to the leading order. See Appendix A for computational details. As a result, the spin-singlet SC and 3$Q$-iCDW repulsively interact each other, and $f_{\text{CDW\&SC}} > 0$ follows from the Cauthy-Schwartz inequality. This implies the opposing behaviour between the two critical temperatures, which is consistent with the tendency observed in experiments [45–48]. We emphasize that this opposing behavior is quite independent of the pairing symmetry and the conventional CDW order parameter beyond our simple model, since the condition, $f_{\text{CDW\&SC}} > 0$, is robust as long as $u_5 > u_4/2$.

Now one can explain how each iCDW pattern from Eq. (5) favors different pairing symmetry $\{\Delta_\alpha\}$ to minimize $f_{\text{CDW\&SC}}$ below $T_{\text{SC}}$. It is mostly determined by the $u_4$-term in Eq. (10) via the phase relationship between iCDW and SC order parameters (Fig. 4). Here, we assume that Eq. (6) still stabilizes the 3$Q$-iCDW preserving the six-fold rotational symmetry even in the presence of SC order parameters. Substituting Eq. (5) into Eq. (10), we find that each iCDW pattern prefers the following SC order parameters,

$$\begin{aligned}
\text{(i)} \quad & \boldsymbol{\Delta}_{\text{patch}} = \sqrt{\frac{1}{2}}\Big(\boldsymbol{\Delta}_{d_{x^2-y^2}} + i\boldsymbol{\Delta}_{d_{xy}}\Big), \\
\text{(ii)} \quad & \boldsymbol{\Delta}_{\text{patch}} = \boldsymbol{\Delta}_s, \\
\text{(iii)} \quad & \boldsymbol{\Delta}_{\text{patch}} = -\frac{1}{3}\boldsymbol{\Delta}_s + \frac{2\sqrt{2}}{3}\boldsymbol{\Delta}_{d_{x^2-y^2}}, \\
\text{(iv)} \quad & \boldsymbol{\Delta}_{\text{patch}} = \Big(\frac{1}{3} + i\sqrt{\frac{1}{3}}\Big)\boldsymbol{\Delta}_s + \Big(\frac{\sqrt{2}}{3} - i\sqrt{\frac{1}{6}}\Big)\boldsymbol{\Delta}_{d_{x^2-y^2}} - \sqrt{\frac{1}{6}}\boldsymbol{\Delta}_{d_{xy}}, \quad (12)
\end{aligned}$$

where we omit the overall amplitudes. Only the $\mathcal{T}$-symmetric solution (ii) favors the pure $s$-wave pairing while the others favour mixtures of the order parameters shown in Eq. (11), namely, (i) $(d + id)$- (iii) $(s + d_{x^2-y^2})$- (iv) $(s + d + id)$-waves. Nevertheless, the amplitudes of the order parameters at all patches are finite and equal to each other, $|\Delta_1| = |\Delta_2| = |\Delta_3|$.

## 5 Experimental implications

We suggest some specific experiments for the detection of iCDW and SC discussed above. We focus on the cases (i) and (iv) in the following and other cases are discussed in Appendix B. First, we consider the STM experiment measuring the local density of states (LDOS) $\rho_\alpha(\mathbf{r}, \omega)$ in real space, where $\mathbf{r}, \alpha$ and $\omega$ represent the location of the unit cell, sublattice, and energy, respectively [19, 49, 50].

For case (i), the STM image is expected to follow the $C_3$-rotation symmetry as shown in Fig. 5. However, in case (iv), the $C_3$-rotation is not preserved, whose symmetry breaking pattern is

demonstrated in Fig. 5. The spectral function is also plotted in Fig. 6, which can be measured in the angle-resolved photoemission spectroscopy (ARPES) experiment [18,51]. With an open boundary condition, the spectral function of the case (i) manifests the topological edge state in the presence of both CDW ($|\Phi_{\alpha\sigma}| = 0.3$) and SC ($|\Delta_\alpha| = 0.1$).

Secondly, the oscillation patterns of the Josephson current as a function of magnetic flux (Fraunhofer pattern) in the corner junction can be used to distinguish different pairing symmetries. The stacked junction between the $s$-wave SC and the kagome SC with (i) ($d+id$) - or (iv) ($s+d+id$) - pairing is schematically illustrated in Fig. 6. For case (i), the perfect cancellation of the critical current in zero field $H_{\text{ext}} = 0$ is a result of the destructive interference at the $\pi/2$-rotated interfaces, which is the common feature of the $d$-wave superconductor [52]. In Appendix C, we also plot the critical current and oscillation patterns in the flat junction. Whereas, for case (iv), the $s$-wave component gives rise to a constructive interference in zero magnetic field, leading to the peak at $\Phi_B = 0$.

# 6 Discussion

Explicit derivation and analysis of the Landau-Ginzburg theory enabled us to conclude that there exist four possible spin-dependent orbital current orders and each one of them may uniquely coexist with one of four possible superconducting order parameters. The relationship between the orbital current order and superconductivity is largely based on symmetry considerations and hence it is robust against various microscopic details. The identification of the orbital current order would imply the presence of a particular kind of superconducting order and vice versa. The proposed experiments such as the STM, ARPES, and Josephson junction tunneling would provide non-trivial checks on our studies.

In recent high pressure experiments, it was shown that the suppression of CDW leads to the enhanced superconductivity upon increasing pressure and a second superconducting dome appears at even higher pressures [14,47,48,50,53–56]. It will be interesting to establish precise relationship between the onset temperature of the anomalous Hall effect and superconductivity as well as CDW in pressurized systems. If the corresponding orbital current order can be identified at different pressure regimes, it may provide an important clue as to whether the same or different kinds of superconducting states arise as a function of hydrostatic pressure.

In the current work, we have not explicitly taken into account the conventional CDW (or the real part of the complex CDW order parameter) for simplicity. The onset temperatures of

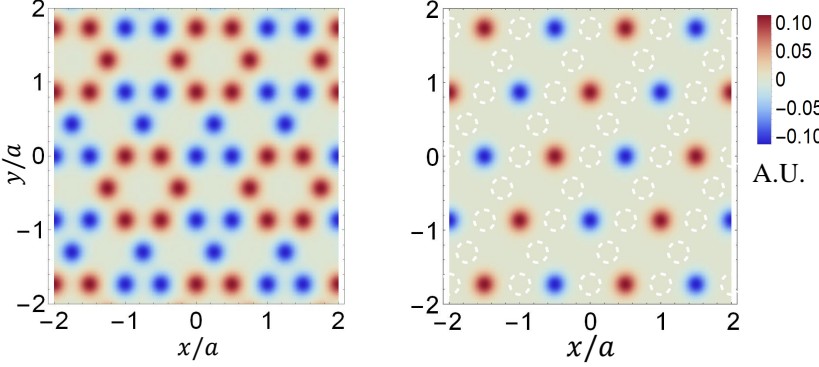

Figure 5: LDOS $\tilde{\rho}_\alpha(\mathbf{r}, \omega = |\Delta_\alpha|)$ measured from the average background (relevant to STM images) for cases (i) left panel and (iv) right panel respectively. Dashed circles mark underlying lattice sites.

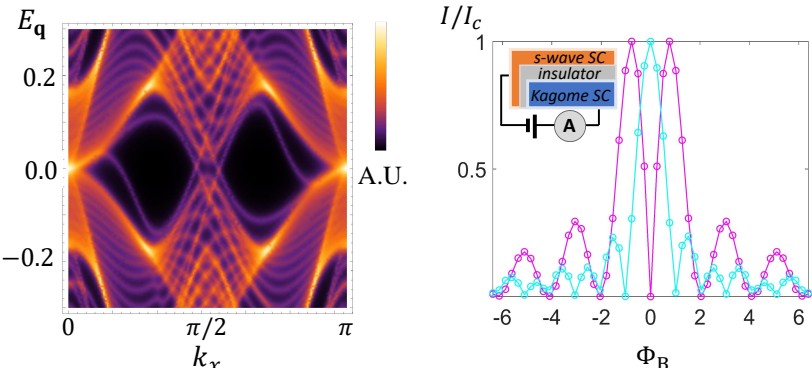

Figure 6: Left panel: Spectral function (relevant to ARPES data) of case (i) $|\Phi_{\alpha\sigma}| = 0.3, |\Delta_\alpha| = 0.1$. Right panel: The Fraunhofer pattern on the conner Josephson junction for case (i) (purple curve), case (iv) (cyan curve) in the presence of (in-plane) magnetic flux $\Phi_B$.

these orders may change, for example, under high pressure or with other external perturbations. We expect that triplet superconductivity such as $f$-wave order might be also significant by tuning microscopic parameters on our simplified model. In addition, the presence of the conventional CDW or multiple vHSs would diversify the kinds of symmetry breaking orders. For realistic applications to the V-based kagome metals, the multi-orbital nature of the vHS needs to be taken into account. These all possibilities and their relation to superconductivity would be interesting questions to explore in future research.

# Acknowledgements

HJY, HSK and SBL are supported by National Research Foundation Grant (NRF-2020R1A4A3079707, NRF grand 2021R1A2C1093060). MYJ and MJH are supported by the National Research Foundation of Korea (NRF) grant funded by the Korea government (MSIT) (No. 2021R1A2C1009303 and No. 2018M3D1A1058754). MYJ and MJH are supported by the KAIST Grand Challenge 30 Project (KC30) in 2021 funded by the Ministry of Science and ICT of Korea and KAIST (N11210105). YBK is supported by the NSERC of Canada and the Center for Quantum Materials at the University of Toronto.

# A   Landau-Ginzburg free energy

In this section, we derive the total LG free energy of complex charge density wave (CDW), $\Phi_{\alpha\sigma}$ and superconducting (SC) order parameters, $\Delta_\alpha$. The real and imaginary parts of $\Phi_{\alpha\sigma}$ correspond to conventional CDW and orbital current respectively. Here, all relevant coefficients of GL free energy in Eqs. (6), (9) and (10) are explicitly evaluated for the perfect nesting, Eq. (2). One can look for Ref. [44] for the similar calculations on the hexagonal lattice.

The total free energy density is

$$
\begin{aligned}
f_{\text{total}} &= \sum_{\alpha\sigma} \frac{1}{2|I_{\text{rCDW}}|}[\text{Re}\Phi_{\alpha\sigma}]^2 + \sum_{\alpha\sigma} \frac{1}{2|I_{\text{iCDW}}|}[\text{Im}\Phi_{\alpha\sigma}]^2 + \frac{1}{2|I_{\text{sSC}}|}|\boldsymbol{\Delta}_s \cdot \boldsymbol{\Delta}_{\text{patch}}^T|^2 \\
&\quad + \frac{1}{2|I_{\text{dSC}}|}\left(|\boldsymbol{\Delta}_{d_{x^2-y^2}} \cdot \boldsymbol{\Delta}_{\text{patch}}^T|^2 + |\boldsymbol{\Delta}_{d_{xy}} \cdot \boldsymbol{\Delta}_{\text{patch}}^T|^2\right) - \frac{1}{\Lambda^2\beta}\text{Tr}\log\left[-\mathcal{G}_k^{-1}\right], \quad (13)
\end{aligned}
$$

where $\Lambda$ and $\beta$ are the linear patch size and inverse temperature respectively. Here, $\alpha \in \{1,2,3\}$ and $\sigma \in \{\uparrow,\downarrow\}$ are patch and spin indices and $I_{\text{op}}$ are the interaction strength for individual order parameters, $\Phi_{\alpha\sigma}$ and $\mathbf{\Delta}_{\text{patch}} = (\Delta_1,\Delta_2,\Delta_3)$. The irreducible channels of SC are [40]

$$\mathbf{\Delta}_s = \sqrt{\frac{1}{3}}(1,1,1), \qquad \mathbf{\Delta}_{d_{x^2-y^2}} = \sqrt{\frac{2}{3}}\left(1,-\frac{1}{2},-\frac{1}{2}\right), \qquad \mathbf{\Delta}_{d_{xy}} = \sqrt{\frac{1}{2}}(0,1,-1). \quad (14)$$

Also $-\mathcal{G}_k^{-1} = -(\mathcal{G}_k^{(0)})^{-1} + \mathcal{V}$ is the full single particle Green function where

$$\mathcal{G}_k^{(0)} = \text{diag}(G_{e1},G_{e2},G_{e3},G_{h1},G_{h2},G_{h3}), \qquad G_{e\alpha(h\alpha)}(i\omega_n,\mathbf{k}) = \frac{1}{i\omega_n \mp \epsilon_{\alpha,\pm\mathbf{k}}}, \quad (15)$$

are bare electron/hole Green functions at the momentum $\mathbf{k}$ and fermion Mastubara frequency $i\omega_n$. Also,

$$\mathcal{V} = \mathcal{V}_{\text{CDW}} + \mathcal{V}_{\text{SC}} = \begin{pmatrix} \tilde{\mathcal{V}}_{\text{CDW}}[\Phi_{\alpha\uparrow}] & \tilde{\mathcal{V}}_{\text{SC}}[\Delta_\alpha] \\ \tilde{\mathcal{V}}_{\text{SC}}^\dagger[\Delta_\alpha] & -\tilde{\mathcal{V}}_{\text{CDW}}^T[\Phi_{\alpha\downarrow}] \end{pmatrix},$$

$$\tilde{\mathcal{V}}_{\text{CDW}}[\Phi_{\alpha\sigma}] = \begin{pmatrix} 0 & \Phi_{3\sigma} & \Phi_{2\sigma}^* \\ \Phi_{3\sigma}^* & 0 & \Phi_{1\sigma} \\ \Phi_{2\sigma} & \Phi_{1\sigma}^* & 0 \end{pmatrix}, \qquad \tilde{\mathcal{V}}_{\text{SC}}[\Delta_\alpha] = \begin{pmatrix} \Delta_1 & 0 & 0 \\ 0 & \Delta_2 & 0 \\ 0 & 0 & \Delta_3 \end{pmatrix}, \quad (16)$$

are decoupled CDW/SC interactions on the basis of $\psi_k = (\psi_{k\uparrow}, (\psi_{-k\downarrow}^\dagger)^T)^T$ with the electronic field $\psi_{k\sigma} = (c_{1k\sigma}, c_{2k\sigma}, c_{3k\sigma})$. Then, Eq. (13) becomes

$$f_{\text{total}} = f_{\text{CDW}} + f_{\text{SC}} + f_{\text{CDW\&SC}}, \qquad f_{\text{CDW}} = f_{\text{CDW}}^\uparrow + f_{\text{CDW}}^\downarrow, \quad (17)$$

$$\begin{aligned} f_{\text{CDW}}^\sigma[\Phi_{\alpha\sigma}] = & \left(\frac{1}{2|I_{\text{rCDW}}|} + \chi_{\text{CDW}}\right)\left(\sum_\alpha [\text{Re}\Phi_{\alpha\sigma}]^2\right) + \left(\frac{1}{2|I_{\text{iCDW}}|} + \chi_{\text{CDW}}\right)\left(\sum_\alpha [\text{Im}\Phi_{\alpha\sigma}]^2\right) \\ & + r\left(\Phi_{1\sigma}\Phi_{2\sigma}\Phi_{3\sigma} + \text{c.c.}\right) + \frac{1}{2}u_1\sum_\alpha |\Phi_{\alpha\sigma}|^4 + u_2\sum_{\alpha<\beta}|\Phi_{\alpha\sigma}|^2|\Phi_{\beta\sigma}|^2, \end{aligned} \quad (18)$$

$$\begin{aligned} f_{\text{SC}}[\Delta_\alpha] = & \left(\frac{1}{2|I_{\text{sSC}}|} + \chi_{\text{SC}}\right)|\mathbf{\Delta}_s \cdot \mathbf{\Delta}_{\text{patch}}^T|^2 \\ & + \left(\frac{1}{2|I_{\text{dSC}}|} + \chi_{\text{SC}}\right)\left(|\mathbf{\Delta}_{d_{x^2-y^2}} \cdot \mathbf{\Delta}_{\text{patch}}^T|^2 + |\mathbf{\Delta}_{d_{xy}} \cdot \mathbf{\Delta}_{\text{patch}}^T|^2\right) + \frac{1}{2}u_3\sum_\alpha |\Delta_\alpha|^4, \end{aligned} \quad (19)$$

$$\begin{aligned} f_{\text{CDW\&SC}}[\Phi_{\alpha\sigma}, \Delta_\alpha] = & -u_4\left(\Phi_{1\uparrow}\Phi_{1\downarrow}\Delta_2^*\Delta_3 + \Phi_{2\uparrow}\Phi_{2\downarrow}\Delta_3^*\Delta_1 + \Phi_{3\uparrow}\Phi_{3\downarrow}\Delta_1^*\Delta_2 + \text{c.c.}\right) \\ & + \frac{1}{2}u_5\sum_{\alpha\beta\gamma}|\epsilon_{\alpha\beta\gamma}|\left(|\Phi_{\alpha\uparrow}|^2 + |\Phi_{\alpha\downarrow}|^2\right)\left(|\Delta_\beta|^2 + |\Delta_\gamma|^2\right), \end{aligned} \quad (20)$$

with coefficients,

$$\chi_{\text{CDW}} = \frac{1}{\Lambda^2\beta}\text{Tr}\left[G_{e1}G_{e2}\right], \qquad \chi_{\text{SC}} = \frac{1}{\Lambda^2\beta}\text{Tr}\left[G_{e1}G_{h1}\right], \qquad r = \frac{1}{\Lambda^2\beta}\text{Tr}\left[G_{e1}G_{e2}G_{e3}\right],$$

$$u_1 = \frac{1}{\Lambda^2\beta}\text{Tr}\left[G_{e1}^2 G_{e2}^2\right], \qquad u_2 = \frac{1}{\Lambda^2\beta}\text{Tr}\left[G_{e1}^2 G_{e2}G_{e3}\right], \qquad u_3 = \frac{1}{\Lambda^2\beta}\text{Tr}\left[G_{e1}^2 G_{h1}^2\right],$$

$$u_4 = \frac{1}{\Lambda^2\beta}\text{Tr}\left[G_{e1}G_{e2}G_{h1}G_{h2}\right], \qquad u_5 = \frac{1}{\Lambda^2\beta}\text{Tr}\left[G_{e1}^2 G_{e2}G_{h1}\right]. \quad (21)$$

In the main text, we focus on the orbital current and its interplay with SC order parameters by setting $\mathrm{Re}\Phi_{\alpha\sigma} = 0$ and $I_{sSC} \approx I_{dSC}$.

The LG expansions, Eqs. (17)-(20) are determined by the symmetry properties of the order parameters $\Phi_{\alpha\sigma}$ and $\Delta_\alpha$ while the coefficients Eq. (21) depend on the material and control parameters. Now, we calculate all free energy coefficients, Eq. (21) for the dispersion, Eq. (2). After the variable transformation, $a = k_x + \sqrt{3}k_y$, $b = k_x - \sqrt{3}k_y$ ($|a|, |b| < \Lambda$), Eq. (2) becomes

$$\epsilon_1 = t'a(a+b) - \mu, \qquad \epsilon_2 = -t'ab - \mu, \qquad \epsilon_3 = t'b(a+b) - \mu, \tag{22}$$

where $t' = t/4$ is rescaled and the chemical potential $\mu \simeq 0$ is kept. Our calculation considers the regime $\mu \ll T \ll t\Lambda^2$ to obtain the leading behaviour as a function of $\mu/T$ and $T/t\Lambda^2$.

$$
\begin{aligned}
\chi_{\mathrm{CDW}} &= \frac{1}{\Lambda^2\beta}\mathrm{Tr}\big[G_{e1}G_{e2}\big] = T\sum_n \int_\Lambda \frac{d^2k}{(2\pi)^2}\Big(\frac{1}{i\omega_n - \epsilon_1}\Big)\Big(\frac{1}{i\omega_n - \epsilon_2}\Big) \\
&= \frac{T}{t'}\sum_n \int_{-\sqrt{t'\Lambda}}^{\sqrt{t'\Lambda}} \frac{da\,db}{(2\pi)^2 2\sqrt{3}}\Big(\frac{1}{i\omega_n - a(a+b) + \mu}\Big)\Big(\frac{1}{i\omega_n + ab + \mu}\Big).
\end{aligned}
\tag{23}
$$

After the contour integral $\int db$ along the upper (or lower) half plane, the integrand depends on the Matsubara frequency sign, $\mathrm{sgn}(\omega_n)$.

$$
\begin{aligned}
\chi_{\mathrm{CDW}} &= \frac{T}{(2\pi)^2 2\sqrt{3}t'}\sum_n \int_{-\sqrt{t'\Lambda}}^{\sqrt{t'\Lambda}} da\, \frac{2\pi i\,\mathrm{sgn}(a)\mathrm{sgn}(\omega_n)}{a(a^2 - 2\mu - 2i\omega_n)} \\
&= \frac{T}{(2\pi)^2 2\sqrt{3}t'}\sum_{\omega_n > 0} \int_{-\sqrt{t'\Lambda}}^{\sqrt{t'\Lambda}} da\, \frac{1}{|a|}\frac{2\pi i\cdot 4i\omega_n}{(a^2 - 2\mu)^2 + 4\omega_n^2} \\
&\approx \frac{T}{(2\pi)^2 2\sqrt{3}t'}\sum_{\omega_n > 0} \int_{-\sqrt{t'\Lambda}}^{\sqrt{t'\Lambda}} da\, \frac{-2\pi}{|a|}\Big(\frac{1}{\omega_n} - \frac{\mu^2}{\omega_n^3}\Big) \\
&\approx -\frac{1}{(2\pi)^2 2\sqrt{3}t'}\Big[\Big(\log\frac{t'\Lambda^2}{\pi T}\Big)^2 - \frac{\mu^2}{(2\pi T)^2}\Big(8.41\log\Big(\frac{t'\Lambda^2}{\pi T}\Big) + 5.28\Big)\Big].
\end{aligned}
\tag{24}
$$

Here, the first term is the most dominant for $\mu \ll T \ll t\Lambda^2$. Similarly,

$$
\begin{aligned}
\chi_{\mathrm{SC}} &= \frac{1}{\Lambda^2\beta}\mathrm{Tr}\big[G_{e1}G_{h1}\big] = T\sum_n \int_\Lambda \frac{d^2k}{(2\pi)^2}\Big(\frac{1}{i\omega_n - \epsilon_1}\Big)\Big(\frac{1}{i\omega_n + \epsilon_1}\Big) \\
&= \frac{T}{t'}\sum_n \int_{-\sqrt{t'\Lambda}}^{\sqrt{t'\Lambda}} \frac{da\,db}{(2\pi)^2 2\sqrt{3}}\Big(\frac{1}{i\omega_n + ab + \mu}\Big)\Big(\frac{1}{i\omega_n - ab - \mu}\Big) \\
&= \frac{T}{(2\pi)^2 2\sqrt{3}t'}\sum_n \int_{-\sqrt{t'\Lambda}}^{\sqrt{t'\Lambda}} da\, \frac{-2\pi i\,\mathrm{sgn}(a)\mathrm{sgn}(\omega_n)}{a\cdot 2i\omega_n} \approx -\frac{1}{(2\pi)^2 2\sqrt{3}t'}\Big(\log\frac{t'\Lambda^2}{\pi T}\Big)^2,
\end{aligned}
\tag{25}
$$

up to the logarithmic accuracy. For a conventional vHS, both the finite momentum CDW and zero-momentum SC susceptibilities diverge as squares of logarithm [40], which results in $T_{\mathrm{CDW}} \sim t\exp\big[-\frac{A}{\sqrt{|I_{\mathrm{CDW}}|}}\big]$, $T_{\mathrm{SC}} \sim t\exp\big[-\frac{B}{\sqrt{|I_{\mathrm{SC}}|}}\big]$ for non-universal constants $A, B$. The cubic coefficient $r$ is,

$$r = \frac{1}{\Lambda^2\beta}\mathrm{Tr}\big[G_{e1}G_{e2}G_{e3}\big] = T\sum_n \int_\Lambda \frac{d^2k}{(2\pi)^2}\Big(\frac{1}{i\omega_n - \epsilon_1}\Big)\Big(\frac{1}{i\omega_n - \epsilon_2}\Big)\Big(\frac{1}{i\omega_n - \epsilon_3}\Big)$$

$$= \frac{T}{t'}\sum_n \int_{-\sqrt{t'\Lambda}}^{\sqrt{t'\Lambda}} \frac{da\,db}{(2\pi)^2 2\sqrt{3}}\Big(\frac{1}{i\omega_n - a(a+b)+\mu}\Big) \tag{26}$$

$$\times \Big(\frac{1}{i\omega_n + ab + \mu}\Big)\Big(\frac{1}{i\omega_n - b(a+b)+\mu}\Big)$$

$$\approx \frac{T}{(2\pi)^2 2\sqrt{3}t'}\sum_{\omega_n > 0}\frac{1}{|\omega_n|^2} 2\mathrm{Re}\int_{-\sqrt{\frac{t'}{\pi T}\Lambda}}^{\sqrt{\frac{t'}{\pi T}\Lambda}} da\,db \Big(\frac{1}{i - a(a+b)+\frac{\mu}{\pi T}}\Big)$$

$$\times \Big(\frac{1}{i + ab + \frac{\mu}{\pi T}}\Big)\Big(\frac{1}{i - b(a+b)+\frac{\mu}{\pi T}}\Big)$$

$$= \frac{1}{(2\pi)^4 2\sqrt{3}t'T}\Big(\sum_{n=0}^{\infty}\frac{1}{(n+\frac{1}{2})^2}\Big)\tilde{r}\Big(\sqrt{\frac{t'}{\pi T}}\Lambda, \frac{\mu}{\pi T}\Big) \approx \frac{4.93}{32\pi^4\sqrt{3}t'T}\tilde{r}\Big(\sqrt{\frac{t'}{\pi T}}\Lambda, \frac{\mu}{\pi T}\Big).$$

Here, $\tilde{r}(x,y)$ is a dimensionless integral with $\tilde{r}(\infty, 0) = 13.1$. It is remarkable that the cubic coefficient sign $r$ is reversed under $t \to -t$ (or $\epsilon_\alpha \to -\epsilon_\alpha$).

$$u_1 = \frac{1}{\Lambda^2\beta}\mathrm{Tr}\Big[G_{e1}^2 G_{e2}^2\Big] = T\sum_n \int_\Lambda \frac{d^2k}{(2\pi)^2}\Big(\frac{1}{i\omega_n - \epsilon_1}\Big)^2\Big(\frac{1}{i\omega_n - \epsilon_2}\Big)^2$$

$$= \frac{T}{t'}\sum_n \int_{-\sqrt{t'\Lambda}}^{\sqrt{t'\Lambda}}\frac{da\,db}{(2\pi)^2 2\sqrt{3}} \tag{27}$$

$$\times \frac{\partial}{\partial\mu_1}\frac{\partial}{\partial\mu_2}\Big[\frac{1}{i\omega_n - a(a+b)+\mu_1}\frac{1}{i\omega_n + ab + \mu_2}\Big]\Big|_{\mu_1=\mu_2=\mu}. \tag{28}$$

After the contour integral and the differentiation ($\partial_{\mu_1}\partial_{\mu_2}$),

$$u_1 = \frac{T}{(2\pi)^2 2\sqrt{3}t'}\sum_n \int_{-\sqrt{t'\Lambda}}^{\sqrt{t'\Lambda}} da \frac{2\cdot 2\pi i\,\mathrm{sgn}(a)\mathrm{sgn}(\omega_n)}{-a(-a^2 + 2\mu + 2i\omega_n)^3}$$

$$= \frac{T}{(2\pi)^2 2\sqrt{3}t'}\sum_{\omega_n>0}\int_{-\sqrt{t'\Lambda}}^{\sqrt{t'\Lambda}} da\frac{4\pi i}{|a|}\frac{12i(a^2-2\mu)^2\omega_n - 16i\omega_n^3}{((a^2-2\mu)^2 + 4\omega_n^2)^3}$$

$$\approx \frac{T}{(2\pi)^2 2\sqrt{3}t'}\sum_{\omega_n>0}\int_{\sqrt{\omega_n}}^{\sqrt{t'\Lambda}}\frac{2\pi}{a}\Big(\frac{1}{\omega_n^3}-\frac{3\mu^2}{\omega_n^5}\Big) = \frac{T}{8\pi\sqrt{3}t'}\sum_{\omega_n>0}\log\Big(\frac{t'\Lambda^2}{\omega_n}\Big)\Big(\frac{1}{\omega_n^3}-\frac{3\mu^2}{\omega_n^5}\Big)$$

$$\approx \frac{1}{64\pi^4\sqrt{3}t'T^2}\Big[8.41\log\Big(\frac{t'\Lambda^2}{2\pi T}\Big)+5.28\Big]$$

$$-\frac{3\mu^2}{256\pi^6\sqrt{3}t'T^4}\Big[32.14\log\Big(\frac{t'\Lambda^2}{2\pi T}\Big)+22.11\Big], \tag{29}$$

up to the logarithmic accuracy.

$$u_2 = \frac{1}{\Lambda^2\beta}\mathrm{Tr}\Big[G_{e1}^2 G_{e2}G_{e3}\Big] = T\sum_n \int_\Lambda \frac{d^2k}{(2\pi)^2}\Big(\frac{1}{i\omega_n - \epsilon_1}\Big)^2\Big(\frac{1}{i\omega_n-\epsilon_2}\Big)\Big(\frac{1}{i\omega_n-\epsilon_3}\Big)$$

$$= \frac{T}{t'}\sum_n\int_{-\sqrt{t'\Lambda}}^{\sqrt{t'\Lambda}}\frac{da\,db}{(2\pi)^2 2\sqrt{3}}\Big(\frac{1}{i\omega_n - a(a+b)+\mu}\Big)^2$$

$$\approx \frac{T}{(2\pi)^2 2\sqrt{3}t'} \sum_{\omega_n>0} \frac{1}{|\omega_n|^3} 2\mathrm{Re} \int_{-\sqrt{\frac{t'}{\pi T}}\Lambda}^{\sqrt{\frac{t'}{\pi T}}\Lambda} dadb \Big( \frac{1}{i-a(a+b)+\frac{\mu}{\pi T}} \Big)^2 \Big( \frac{1}{i+ab+\frac{\mu}{\pi T}} \Big)$$

$$\times \Big( \frac{1}{i-b(a+b)+\frac{\mu}{\pi T}} \Big) = \frac{1}{(2\pi)^5 2\sqrt{3}t'T^2} \Big( \sum_{n=0}^{\infty} \frac{1}{(n+\frac{1}{2})^3} \Big) \tilde{u}_2 \Big( \sqrt{\frac{t'}{\pi T}}\Lambda, \frac{\mu}{\pi T} \Big)$$

$$\approx \frac{8.41}{64\pi^5 \sqrt{3}t'T^2} \tilde{u}_2 \Big( \sqrt{\frac{t'}{\pi T}}\Lambda, \frac{\mu}{\pi T} \Big), \tag{30}$$

where $\tilde{u}_2(x,y)$ is the dimensionless integral with $\tilde{u}_2(\infty,0)=5.81$. With leading behaviours, Eqs. (29) and (30) reduces to Eq. (7) in the main text.

Likewise, $u_3, u_4, u_5$ are calculated,

$$
\begin{aligned}
u_3 &= \frac{1}{\Lambda^2 \beta} \mathrm{Tr}\Big[ G_{e1}^2 G_{h1}^2 \Big] = T \sum_n \int_\Lambda \frac{d^2k}{(2\pi)^2} \Big( \frac{1}{i\omega_n-\epsilon_1} \Big)^2 \Big( \frac{1}{i\omega_n+\epsilon_1} \Big)^2 \\
&= \frac{T}{t'} \sum_n \int_{-\sqrt{t'}\Lambda}^{\sqrt{t'}\Lambda} \frac{dadb}{(2\pi)^2 2\sqrt{3}} \Big( \frac{\partial}{\partial\mu_1} \frac{\partial}{\partial\mu_2} \frac{1}{i\omega_n+ab+\mu_1} \frac{1}{i\omega_n-ab-\mu_2} \Big) \Big|_{\mu_1=\mu_2=\mu} \\
&= \frac{T}{(2\pi)^2 2\sqrt{3}t'} \sum_n \int_{-\sqrt{t'}\Lambda}^{\sqrt{t'}\Lambda} da \frac{2\pi i}{a} \frac{\mathrm{sgn}(a)\mathrm{sgn}(\omega_n)}{2i\omega_n} \\
&\approx \frac{1}{(2\pi)^4 4\sqrt{3}t'T^2} \sum_{n=0}^{\infty} \frac{1}{(n+\frac{1}{2})^3} \log\Big( \frac{t\Lambda^2}{2\pi T(n+\frac{1}{2})} \Big) \\
&= \frac{1}{64\pi^4 \sqrt{3}t'T^2} \Big[ 8.41 \log\Big( \frac{t'\Lambda^2}{2\pi T} \Big) + 5.28 \Big],
\end{aligned}
\tag{31}
$$

$$
\begin{aligned}
u_4 &= \frac{1}{\Lambda^2 \beta} \mathrm{Tr}\Big[ G_{e1} G_{e2} G_{h1} G_{h2} \Big] \\
&= T \sum_n \int_\Lambda \frac{d^2k}{(2\pi)^2} \Big( \frac{1}{i\omega_n-\epsilon_1} \Big) \Big( \frac{1}{i\omega_n-\epsilon_2} \Big) \Big( \frac{1}{i\omega_n+\epsilon_1} \Big) \Big( \frac{1}{i\omega_n+\epsilon_2} \Big) \\
&= \frac{T}{t'} \sum_n \int_{-\sqrt{t'}\Lambda}^{\sqrt{t'}\Lambda} \frac{dadb}{(2\pi)^2 2\sqrt{3}} \Big( \frac{1}{i\omega_n-a(a+b)+\mu} \Big) \Big( \frac{1}{i\omega_n+ab+\mu} \Big) \\
&\quad \times \Big( \frac{1}{i\omega_n+a(a+b)-\mu} \Big) \Big( \frac{1}{i\omega_n-ab-\mu} \Big) \\
&= \frac{T}{(2\pi)^2 2\sqrt{3}t'} \sum_n \int_{-\sqrt{t'}\Lambda}^{\sqrt{t'}\Lambda} da \frac{2\pi i \mathrm{sgn}(a)\mathrm{sgn}(\omega_n)}{2ia\omega_n} \Big( \frac{2}{(a^2-2\mu)^2+4\omega_n^2} \Big) \\
&\approx \frac{T}{(2\pi)^2 2\sqrt{3}t'} \pi \sum_{\omega_n>0} \int da \frac{1}{|a|} \Big( \frac{1}{\omega_n^3} - \frac{\mu^2}{\omega_n^5} \Big) \\
&\approx \frac{1}{64\pi^4 \sqrt{3}t'T^2} \Big[ 8.41 \log\Big( \frac{t'\Lambda^2}{2\pi T} \Big) + 5.28 \Big] \\
&\quad - \frac{\mu^2}{256\pi^6 \sqrt{3}t'T^4} \Big[ 32.14 \log\Big( \frac{t'\Lambda^2}{2\pi T} \Big) + 22.11 \Big],
\end{aligned}
\tag{32}
$$

$$u_5 = \frac{1}{\Lambda^2 \beta} \mathrm{Tr}\Big[ G_{e1}^2 G_{e2} G_{h1} \Big] = T \sum_n \int_\Lambda \frac{d^2k}{(2\pi)^2} \Big( \frac{1}{i\omega_n-\epsilon_1} \Big)^2 \Big( \frac{1}{i\omega_n-\epsilon_2} \Big) \Big( \frac{1}{i\omega_n+\epsilon_1} \Big)$$

$$
\begin{aligned}
&= \quad \frac{T}{(2\pi)^2 2\sqrt{3}t'} \sum_n \int da \frac{2\pi i\,\mathrm{sgn}(a)\,\mathrm{sgn}(\omega_n)}{4a\omega_n^2} \frac{-(a^2 - 2\mu - 4i\omega_n)}{(a^2 - 2\mu - 2i\omega_n)^2} \\
&\approx \quad \frac{T}{(2\pi)^2 2\sqrt{3}t'} \pi \sum_{\omega_n>0} \int da \frac{1}{|a|}\Big(\frac{1}{\omega_n^3} - \frac{2\mu^2}{\omega_n^5}\Big) \\
&\approx \quad \frac{1}{64\pi^4\sqrt{3}t'T^2}\Big[8.41\log\Big(\frac{t'\Lambda^2}{2\pi T}\Big) + 5.28\Big] \\
&\quad - \frac{\mu^2}{128\pi^6\sqrt{3}t'T^4}\Big[32.14\log\Big(\frac{t'\Lambda^2}{2\pi T}\Big) + 22.11\Big].
\end{aligned}
\tag{33}
$$

In the last expressions, Eqs. (29)-(33), the first term is most dominant and the others are subleading, which results in $u_1 \approx u_3 \approx u_4 \approx u_5 \gg u_2$ up to the leading order. To estimate the numerical factors in Eqs. (28)-(33), we refer the following table. With the zeta function value at $z$, $\zeta(z)$,

$$
\sum_{n=0}^{\infty}\frac{1}{(n+\frac{1}{2})^2} = \frac{\pi^2}{2} \approx 4.93\,, \quad \sum_{n=0}^{\infty}\frac{1}{(n+\frac{1}{2})^3} = 7\zeta(3) \approx 8.41\,,
$$

$$
\sum_{n=0}^{\infty}\frac{1}{(n+\frac{1}{2})^5} = 31\zeta(5) \approx 32.14\,,
$$

$$
\sum_{n=0}^{\infty}\frac{1}{(n+\frac{1}{2})^3}\log\Big(n+\frac{1}{2}\Big) = -8\zeta(3)\log 2 - 7\zeta'(3) \approx -5.28\,,
$$

$$
\sum_{n=0}^{\infty}\frac{1}{(n+\frac{1}{2})^5}\log\Big(n+\frac{1}{2}\Big) = -32\zeta(5)\log 2 - 31\zeta'(5) \approx -22.11\,.
\tag{34}
$$

## B  Microscopic Hamiltonian

Although the order parameters $\Phi_{\alpha\sigma}$ and $\Delta_\alpha$ are defined in the patch model, the microscopic Hamiltonian can be established before the approximation close to vHS. Because of the kagome lattice geometry, the eigenstate weight at $\mathbf{M}_{\alpha=1,2,3}$ is composed of the corresponding sublattice $V_{\alpha=1,2,3}$ respectively. As a result, the iCDW of finite momentum $\mathbf{M}_\alpha$ generates the imaginary hopping between nearest-neighbour sites with the angular momentum $l = 1$ [36,37]. For the down-spin electrons, there are independent 4 cases in Eq. (5) whose Peierls phases and the plaquette fluxes are shown in Fig. 2.

In the same manner, SC order parameter, Eq. (8) is the condensate of paired electrons at the same sublattice. Then on-site and the third-neighbour interactions are responsible for the $s$- and $d$-wave pairings respectively. In our calculation, the extended $s$-wave pairings beyond the on-site interaction are not considered. Then, the gap function in the Bogoliubov-de Gennes (BdG) Hamiltonian is

$$
\begin{aligned}
\tilde{\Delta}_{\mathbf{p}} = \quad &\Big[\frac{1}{\sqrt{3}}\Delta_s + \frac{1}{2}\Delta_{d_{x^2-y^2}}\Big(\sqrt{\frac{2}{3}}\cos(\mathbf{p}\cdot(\boldsymbol{a}_2 - \boldsymbol{a}_1)) - \sqrt{\frac{1}{6}}\cos(\mathbf{p}\cdot\boldsymbol{a}_1) - \sqrt{\frac{1}{6}}\cos(\mathbf{p}\cdot\boldsymbol{a}_2)\Big) \\
&+ \frac{1}{2}\Delta_{d_{xy}}\Big(\sqrt{\frac{1}{2}}\cos(\mathbf{p}\cdot\boldsymbol{a}_1) - \sqrt{\frac{1}{2}}\cos(\mathbf{p}\cdot\boldsymbol{a}_2)\Big)\Big]\cdot\boldsymbol{\Delta}_{\mathrm{patch}}^T,
\end{aligned}
\tag{35}
$$

for all three sublattices. Here, $\mathbf{a}_{1,2}$ are the primitive lattice vectors (Fig. 1) and $\mathbf{p}$ is the crystal momentum (not the momentum deviation $\mathbf{k}$ in the patch model). At the vHS, $\Delta_{\mathbf{p}}$ reduces to the SC order parameters in the patch model, i.e. $\tilde{\Delta}_{\mathbf{p}=\mathbf{M}_\alpha} = \Delta_\alpha$. In the presence of iCDW, the unit cell is extended to contain 12 sublattices, and the gap function Eq. (35) is appropriately

modified. In the presence both iCDW and SC order parameters, the spectral functions for the Bogoliubov Hamiltonian are shown in Fig. 7.

## C  Josephson junction

The Josephson junction measures the supercurrent between two superconductors separated by an insulating barrier. The SC Hamiltonians of the patch model Eqs. (15) and (16) are given

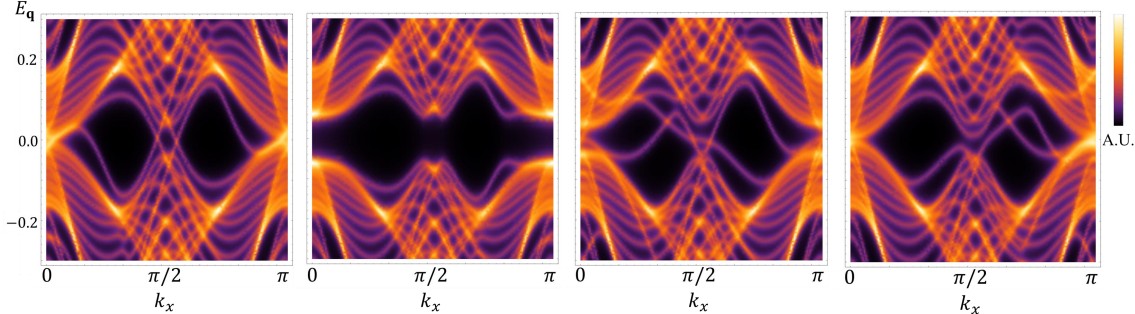

Figure 7: Spectral functions of BdG Hamiltonian, Eq. (35) with an open boundary condition for $(\Phi_{1\downarrow}, \Phi_{2\downarrow}, \Phi_{3\downarrow}) = (i, i, i)$, $(\Phi_{1\downarrow}, \Phi_{2\downarrow}, \Phi_{3\downarrow}) = (-i, -i, -i)$, $(\Phi_{1\downarrow}, \Phi_{2\downarrow}, \Phi_{3\downarrow}) = (-i, i, i)$, and $(\Phi_{1\downarrow}, \Phi_{2\downarrow}, \Phi_{3\downarrow}) = (-i, -i, i)$ (from left to right). Here, the order parameters are $|\Phi_{\alpha\sigma}| = 0.3, |\Delta_\alpha| = 0.1$.

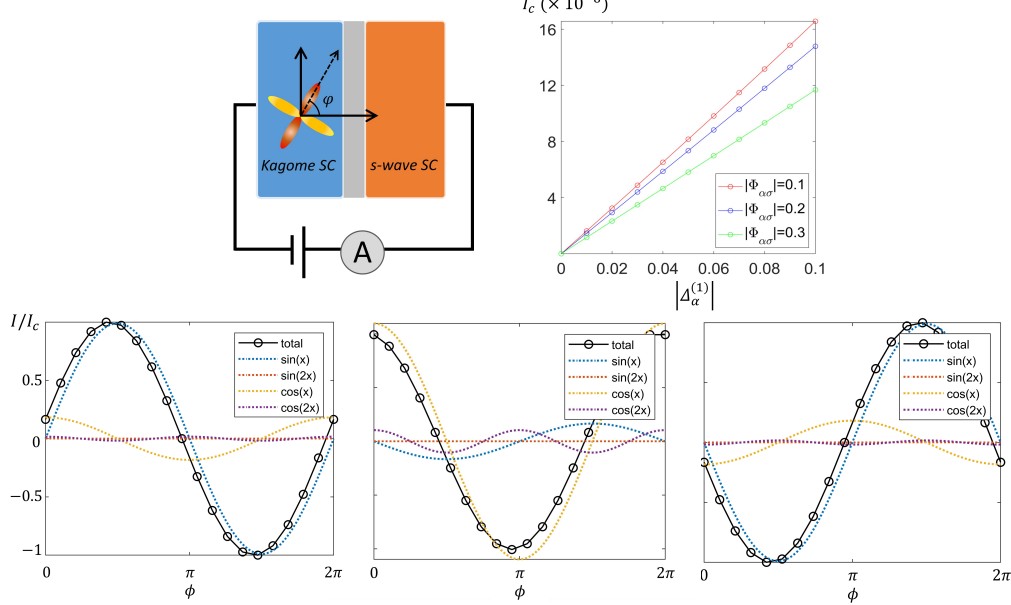

Figure 8: In first row, the schematic illustration of flat junction of kagome SC (case (i)) and the conventional $s$-wave SC separated by an insulating barrier. The magnitude of critical currents as we tuned the iCDW ($\Phi_{\alpha\sigma}$) and SC ($\Delta_\alpha$) order parameters is plotted. In second row, the current-phase relation as we rotate the orientation, $\varphi = 0, \pi/4$ and $\pi/2$ (from left to right) respectively. Here the current is evaluated with order parameters $|\Phi_{\alpha\sigma}| = 0.3, |\Delta_\alpha^{(1)}| = |\Delta_\alpha^{(2)}| = 0.1$, temperature $k_B T = 0.1$, and tunnelling amplitude $t_{\text{tunnel}} = 0.1$ in unit of $t = 1$.

by

$$H_1 \equiv H_1[c_{\alpha\mathbf{k}\sigma}, c_{\alpha\mathbf{k}\sigma}^\dagger, \Phi_{\alpha\sigma}^{(1)}, \Delta_\alpha^{(1)}], \qquad H_2 \equiv H_2[f_{\alpha\mathbf{q}\sigma}, f_{\alpha\mathbf{q}\sigma}^\dagger, \Phi_{\alpha\sigma}^{(2)}, \Delta_\alpha^{(2)}], \tag{36}$$

where the subsystem 1 is the conventional SC with $s$-wave pairing, $\Phi_{\alpha\sigma}^{(1)} = 0, \Delta_{\text{patch}}^{(1)} = |\Delta_\alpha^{(1)}|\Delta_s$ and the subsystem 2 is the kagome SC in which the order parameters $\{\Phi_{\alpha\sigma}^{(2)}, \Delta_\alpha^{(2)}\}$ are taken from one of four cases in Eqs. (5) and (12).

$$H_{\text{tunnel}} = t_{\text{tunnel}} \sum_{\alpha\mathbf{k}\beta\mathbf{q}} \sum_\sigma \left[ c_{\alpha\mathbf{k}\sigma}^\dagger f_{\beta\mathbf{q}\sigma} e^{i\phi/2} + \text{h.c.} \right], \qquad \left( t_{\text{tunnel}} \text{ is real} \right), \tag{37}$$

where $\phi$ is the phase difference of SC order parameters between two subsystems. Then the total Hamiltonian is

$$H_{\text{total}} = H_1 + H_2 + H_{\text{tunnel}}, \tag{38}$$

and the tunnelling current is

$$I_{1\to 2} = \langle \frac{\partial H_{\text{total}}}{\partial \phi} \rangle = \frac{\partial F_{\text{total}}[\phi]}{\partial \phi}, \tag{39}$$

where $\langle ... \rangle$ is the expectation value and $F_{\text{total}}$ is the free energy with respect to the total Hamiltonian.

Since only the SC order parameter of case (i) is the pure $d$-wave pairing among 4 cases in Eq. (12), the phase shift can be observed as the orientation of the flat junction, $\varphi$ rotates. (Fig. 8). The presence of iCDW suppress the magnitude of the critical current, but the current-phase relation ($I/I_c$ versus $\phi$) is apparent. The periodic function, $I(\phi + 2\pi) = I(\phi)$ is expanded as a Fourier series up to the second harmonics. The $\pi$-shift under $\varphi = \pi/2$-rotation is the common feature of $d$-wave SC [52]. In the main text, we also plot the Fraunhofer pattern in the conner Josephson junction to compare cases (i) and (iv).

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
