# Peer review of "Intertwining orbital current order and superconductivity in Kagome metal"

_SciPost Physics, doi:SciPost Phys. Core 6, 008 (2023)_

## Round 2 · Referee Report · Anonymous (Referee 3) · 2022-9-19

Report

The authors have adequately my comments from the previous report and made appropriate changes of the manuscript . I fully support publication of this timely and interesting work in the SciPost journal.

---

## Round 2 · Referee Report · Anonymous (Referee 2) · 2022-9-26

Report

I deem the authors' response fair. However, in my opinion it also demonstrates the limited scope of this manuscript that considers the highly simplified situation of the single-band model with the purely imaginary CDW. These simplifications can not be reasonably justified in AV3Sb5 or any other kagome metal that has been studied experimentally. Even if some experimental implications are given in Section 5, they would not be relevant to any realistic situation. For example, the C3 symmetry of LDOS measured by STM is proposed as one of the fingerprints of the superconducting state (Fig. 5), but in AV3Sb5 both six-fold and three-fold symmetries are already lifted by the real part of the CDW that results in a sizable modulation of the atomic positions / charge density [arXiv:2110.11306; arXiv:2203.15057; PRB 105, 195136 (2022); arXiv:2208.01499]. Therefore, the proposed experiment will not give any information in this case.

On reading through the authors' response, I could not find any statement on how their manuscript meets the acceptance criteria of SciPost Physics. Unless I miss some important message hidden between the lines, I would have a hard time justifying why this paper should appear in the journal that seeks to "detail a groundbreaking discovery, present a breakthrough, or open a new pathway with clear potential for multipronged follow-up work". Therefore, I can recommend publication in SciPost Phys. Core only.

---

## Round 2 · Author Response

Dear editor

We are resubmitting our manuscript "Intertwining orbital current order and superconductivity in Kagome metal". Based on three referee's comments, we have completed the response letter and changed our manuscript for clarification.

---

## Round 2 · List of Changes

List of changes

  1. Introduction On the third paragraph, after the sentence “Considering the spin degrees of freedom, … SC order parameter.”, we add a sentence to emphasize the importance of iCDW order. “It turns out that the orbital current order is closely related to both the non-trivial band topology and unconventional SC order parameters.”

  2. iCDW patterns We move the figs. 5 and 6 from the Appendix to Section 3 for clarity. The Appendix C Edge spectrum is erased and a sentence is added below Eq. (5). “In Fig. 2, we exhibit the Peierls phases and the plaquette fluxes for 4 iCDW patterns.” and at the end of Section 3, a small paragraph is added. “Here, we exhibit the edge spectrum and ... in Fig. 3.”

4.2. Total free energy Starting the subsection, we add a sentence to explain the ansatz considering the intra-pairing (s-, d-wave SC orders) only. “In the perfect nesting, Eq. (2), since the particle-particle susceptibility on the same patch is predominant, we focus on the SC order parameters which pair intra-patch excitations.”

  1. Discussion On the third paragraph, after the sentence “The onset temperatures … other external perturbations.”, we add a sentence for motivate future research by tuning our simplified model. We also add another sentence to emphasize required future work for realistic applications to the Kagome metals. “We expect that triplet superconductivity such as f-wave order might be also significant by tuning microscopic parameters on our simplified model.” “For realistic applications to the V-based Kagome metals, the multi-orbital nature of the vHS needs to be taken into account.”

B. Microscopic Hamiltonian At the end of Appendix B, a sentence was added. “In the presence both iCDW and SC order parameters, the spectral functions for the Bogoliubov Hamiltonian are shown in Fig. 7.”

---

## Editorial Decision

published